# ENT Residents Benefit from a Structured Operation Planning Approach in the Training of Functional Endoscopic Sinus Surgery

**DOI:** 10.3390/medicina57101062

**Published:** 2021-10-04

**Authors:** Sven Becker, Phillipp Gonser, Magnus Haas, Martin Sailer, Matthias F. Froelich, Christian Betz, Hubert Löwenheim, Bernhard Hirt, Wieland H. Sommer, Martin Holderried, Benjamin P. Ernst

**Affiliations:** 1Department of Otorhinolaryngology, Head and Neck Surgery, University Medical Center of Tübingen, 72076 Tübingen, Germany; sven.becker@med.uni-tuebingen.de (S.B.); martin.sailer@med.uni-tuebingen.de (M.S.); hubert.loewenheim@med.uni-tuebingen.de (H.L.); 2Eberhard Karls University Tübingen, 72076 Tübingen, Germany; magnus.haas@student.uni-tuebingen.de; 3Department of Radiology and Nuclear Medicine, University Medical Center Mannheim, 68167 Mannheim, Germany; matthias.froelich@umm.de; 4Department of Otorhinolaryngology, University Medical Center Hamburg-Eppendorf, 20246 Hamburg, Germany; c.betz@uke.de; 5Institute for Clinical Anatomy and Cell Analysis, Eberhard Karls University Tübingen, 72076 Tübingen, Germany; bernhard.hirt@med.uni-tuebingen.de; 6Department of Radiology, LMU University Hospital, 81377 München, Germany; w.sommer@smart-reporting.com; 7Department of Quality Management, Medical and Business Development, University Hospital of Tübingen, 72076 Tübingen, Germany; martin.holderried@med.uni-tuebingen.de; 8Department of Otorhinolaryngology, University Medical Center Bonn, 53127 Bonn, Germany; benjamin.ernst@ukbonn.de

**Keywords:** structured surgical training, medical informatics, functional endoscopic sinus surgery, medical education, computed tomography

## Abstract

*Background and Objectives*: Preoperative planning utilizing computed tomographies (CT) is of utmost importance in functional endoscopic sinus surgery (FESS). Frequently, no uniform documentation and planning structures are available to residents in training. Consequently, overall completeness and quality of operation planning may vary greatly. The objective of the present study was to evaluate the impact of a structured operation planning (SOP) approach on the report quality and user convenience during a 4-day sinus surgery course. *Materials and Methods*: Fifteen participant were requested to plan a FESS procedure based on a CT scan of the paranasal sinuses that exhibited common pathological features, in a conventional manner, using a free text. Afterwards, the participants reevaluated the same scans by means of a specifically designed structured reporting template. Two experienced ENT surgeons assessed the collected conventional operation planning (COP) and SOP methods independently with regard to time requirements, overall quality, and legibility. User convenience data were collected by utilizing visual analogue scales. *Results*: A significantly greater time expenditure was associated with SOPs (183 s vs. 297 s, *p* = 0.0003). Yet, legibility (100% vs. 72%, *p* < 0.0001) and overall completeness (61.3% vs. 22.7%, *p* < 0.0001) of SOPs was significantly superior to COPs. Additionally, description of highly relevant variants in anatomy and pathologies were outlined in greater detail. User convenience data delineated a significant preference for SOPs (VAS 7.9 vs. 6.9, *p* = 0.0185). *Conclusions*: CT-based planning of FESS procedures by residents in training using a structured approach is more time-consuming while producing a superior report quality in terms of detailedness and readability. Consequently, SOP can be considered as a valuable tool in the process of preoperative evaluations, especially within residency.

## 1. Introduction

Over the last decades, functional endoscopic sinus surgery (FESS) has become the standard of care in paranasal sinus surgery with significant efficacy in the treatment of sinonasal diseases [1]. Multiplanar high-resolution computed tomography (CT) not only plays a major role in the diagnostic process, but helps to determine the course of treatment as the images clearly show variants in anatomy and the degree of disease. Moreover, CT scans allow the differentiation of inflammatory, benign, and malignant sinonasal pathologies [2]. In particular, the identification of potentially dangerous anatomical characteristics, such as the depth of the frontal skull base or the course and bony coverage of the optic nerve and internal carotid artery, are of central importance to minimize the risks of the operation [3,4,5,6,7,8,9].

A profound understanding of the sinonasal anatomy and its variations is indispensable to safely perform FESS. Preoperative CT scans help to precisely plan a surgical roadmap that highlights any potentially hazardous anatomical features, and by delineating the degree of disease, prevent unneeded dissections in disease-free parts of the paranasal sinuses [6,7,8,10,11,12]. Furthermore, even though understanding of sinonasal pathologies, surgical approaches as well as radiological imaging have undergone an evolution in both content and structure [13], radiologic reporting has just started to evolve over the course of the past years. Structured reporting (SR) has proven to be an auspicious approach compared to the common practice of free text reporting to standardize the content of reporting and thereby improve the report quality of several diagnostic modalities in otorhinolaryngology that are favored by many physicians [14,15,16,17,18,19,20,21,22,23]. SR templates containing standardized terminology help to reduce the likelihood of missing key structures [11] and, consequently, may be of valuable help to improve surgical operation planning (SOP), especially for younger physicians in clinical routines [24,25].

The purpose of this study was to analyze the effect of SOP on the learning process of FESS. For this purpose, the quality, the time to plan the FESS procedure, and the user convenience of conventional operation planning (COP) using free text reporting in contrast to the structured operation planning of sinus CTs utilizing a specific SR template, were analyzed in the context of a 4-day FESS immersion course at a university medical center.

## 2. Materials and Methods

This study was conducted in accordance with the STROBE guidelines [26]. Due to the design of the study, no approval from the local ethics committee needed to be obtained.

As previously described, the anticipated effect size was used to determine the amount of reports needed for this study. At the significance level of α = 0.05, we set the power to be 80%. Therefore, the necessary number of reports demanded within this study was *n* = 28 (14 COPs and SOPs, respectively).

During a 4-day annual course on FESS at the university medical center in Tübingen, 15 out of 20 (75%) participating physicians agreed to participate in the study. Gender, level of training (resident or consultant), work environment (medical practice, municipal hospital, or university medical center) and individual level of experience in FESS were evaluated using a five-point visual analogue scale (VAS; 1: no experience; 5: highly experienced) (see Table 1).

Within the immersion course, all participating residents received training to create COP using free texts prior to performing the tasks. For the study, the subjects were requested to create free-text-based COPs of a previously unknown CT scan of the paranasal sinuses that exhibited common pathological features of a chronic unilateral sinusitis for preoperative FESS evaluation, as they would do in their own work environment. Afterwards, the participants were asked to repeat this task while using a specifically designed SOP template. To create this template, an established SR system (Smart Reporting GmbH, Munich, Germany, https://www.smart-reporting.com/en/, accessed on 3 October 2021, see Appendix A) was utilized. The template was developed with consideration of the most recent directives for CT-based planning of FESS procedures. A special focus was set to incorporate relevant anatomical structures, adequate terminology, and a wide range of paranasal sinus diseases, both benign and malignant. Additionally, the template was previously validated in routine clinical use [25].

Time expenditure to plan the operations in seconds was recorded for each approach. Subsequently, each participant completed a specifically designed questionnaire rating user convenience and usability by means of a ten-point VAS.

Anonymized COP and SOP reports were independently assessed, by two highly experienced FESS surgeons, in terms of the completeness of the identified critical anatomical structures and the determination of the extent of the procedure, in addition to legibility (five graded scale ranging from 1 = poor to 5 = very good), using a specifically designed evaluation checklist.

Data are reported as mean ± standard deviation (SD). To compare assessments of surgical planning and user convenience evaluations, the Wilcoxon signed-rank test for paired nominal data was used, with a *p*-value of less than 0.05 defined as being statistically significant. All statistical tests were carried out utilizing Prism 9 (GraphPad Software, Inc., San Diego, CA, USA).

## 3. Results

FESS procedures were planned using COP and SOP by otorhinolaryngologists in training as well as by board-certified otorhinolaryngologists, during a 4-day FESS course at the Department of Otorhinolaryngology, Head and Neck Surgery of the University Hospital of Tübingen, using a previously unknown sinus CT with typical pathologies (chronic unilateral sinusitis). Demographic data and levels of experience of the 15 participating otorhinolaryngologists are shown in Table 1.

The time expenditure to plan an operation conventionally was significantly shorter than for the respective structured approach (183 s ± 90 vs. 296 s ± 95, *p* = 0.0003). Still, SOPs exhibited better legibility (100.0% ± 0 vs. 72.0% ± 12.2, *p* < 0.0001) and better overall completeness (95.2% ± 5.3 vs. 32.5% ± 2.4, *p* < 0.0001) in comparison to COP (see Figure 1).

Detailed analysis of completeness findings outlined significantly superior results for SOP compared to COP regarding the nasal septum (95.0% ± 13.5 vs. 50.7% ± 17.7, *p* = 0.001), the middle nasal meatus (100.0% ± 0 vs. 44.0% ± 10.8, *p* < 0.0001), the ethmoidal infundibulum (100.0% ± 0 vs. 36.0% ± 10.8, *p* < 0.0001), the maxillary sinus (88.0% ± 9.8 vs. 40.0% ± 0, *p* < 0.0001), the ethmoidal sinuses (93.3% ± 10.2 vs. 40.0% ± 0, *p* < 0.0001), the sphenoid sinus (97.7% ± 5.7 vs. 45.3% ± 11.5, *p* < 0.0001), and the frontal sinus (94.3% ± 14.6 vs. 42.7% ± 6.8, *p* < 0.0001). Potentially existing masses within the paranasal sinuses (93.3% ± 24.9 vs. 0%, *p* = 0.0001), as well as the depth of the anterior skull base and olfactory fossa (73.3% ± 44.2 vs. 0%, *p* = 0.001), measured using the Keros classification, were only considered using SOP (see Figure 1).

The user convenience analysis, conducted via a VAS-based questionnaire, showed a significant overall preference for SOPs by all participating otorhinolaryngologists (7.9 ± 1.9 vs. 6.9 ± 3.2, *p* = 0.0185). Although detailed analysis for usefulness (8.0 ± 1.2 vs. 7.7 ± 2.9, *p* = 0.97), usability in everyday practice (8.3 ± 1.2 vs. 7.3 ± 2.9, *p* = 0.35), and improvement of the quality of preoperative planning (8.1 ± 1.7 vs. 6.9 ± 3.2, *p* = 0.12) showed a tendency towards SOP, these tendencies were not statistically significant. The same applied to the question of whether SOP can save time in the preoperative workup (6.1 ± 2.8 vs. 5.2 ± 3.0, *p* = 0.4) and if the additional time is justified (8.0 ± 0.9 vs. 7.2 ± 3.2, *p* = 0.53), as well as whether SOP supports unexperienced surgeons in their learning progress (9.2 ± 0.9 vs. 7.3 ± 3.4, *p* = 0.11). A graphical illustration of the results of the questionnaire is provided in Figure 2.

All participants stated that the software for creating SOPs was intuitive, easy to navigate, and easy to learn.

## 4. Discussion

Precise evaluation of preoperative CT scans and step-by-step planning of the FEES procedure is necessary to achieve good operative long-term results on the one hand and to avoid complications due to overseen anatomical variants on the other hand. Although surgical techniques and equipment have undergone tremendous developments, structured operation planning based on sinus CT-scans by the surgeons themselves has so far not been comprehensively investigated [27,28]. This is remarkable because structured, detailed planning of surgical procedures can boost the learning curve and can also increase the confidence of younger physicians in training [29,30].

Therefore, the current study evaluated the time expenditure, legibility, completeness, and user convenience of COP in contrast to SOP. The study population consisted of participants of an annual 4-day FESS course at the Department of Otorhinolaryngology, Head and Neck Surgery of the University Hospital of Tübingen, all of whom had been working in the field of ENT for a minimum of 3 years and estimated their expertise in sinus surgery to be average. All of the participants had already assisted in FESS operations and almost half of them had already performed FESS themselves under supervision. Consequently, a profound knowledge on the reporting of sinus CT scans, and also on the planning of FESS procedures, could be presumed. Furthermore, all participants received a special training in sinus CT reporting as part of the course before enrolling in the study.

The time needed to create COP was less than for SOP (COP 183 s vs. SOP 297 s, *p* < 0.0001). This aspect inversely correlated with the completeness of findings in the participants’ reports, which was higher in SOP (95.2% ± 5.3 vs. 32.5% ± 2.4, *p* < 0.0001).

The increased time expenditure in SOP is concordant with the studies of Sluijter and Ernst et al., which showed a decrease in time-efficiency following the implementation of SR [23,31]. One reason for this finding in our study may be due to the short duration of training of the participants in the usage of the SR template and the underlying software. One aspect that could be observed during the creation of the SOP by the participants was that even though they had already seen the CT scan of the patient, which was necessary to produce the COP, they checked on it again, specifically and more thoroughly, prior to producing the SOP. Classifications of anatomical findings such as the length of the lateral cribriform lamella (Keros classification) were not mentioned by any of the participants when using COP, but by nearly three-quarters of participants when using SOP. This may have had an even greater impact on the time to complete the task. In consequence, SOP took more time but yielded a higher completeness of findings according to the results of our evaluation. The increase in reporting quality has already been shown in several studies and seems to be one of the main advantages in SR over free text reporting [14,15,16,17,19,23,31]. In particular, the decrease in the variability of reports of the same findings is of central interest in the context of quality assurance, scientific data analysis, and medical education. This effect may be explained by the structured procedure in which the reporting physician is guided through the relevant anatomy by the underlying decision tree. In addition, it supports inexperienced physicians through the standardization of language and the description of specific pathologies, which may potentially increase the inter-rater-reliability of SOP [20]. In consequence, the use of SOP may promote information extraction and enhance clinical decision-making. Additionally, there is evidence that a structured approach may reduce interobserver variability [16,17,19]. The COVER survey supports this aspect and states that clinicians welcome the implementation of structured documentation [21]. Additionally, the American Society of Radiology, the Radiological Society of Northern America, as well as the European Society of Radiology have started action groups to advance SR over the past years [18,32,33].

The template utilized in this study was developed, with the collaboration of otorhinolaryngologists and radiologists with high levels of expertise in the diagnosis and treatment of sinonasal diseases, to promote user-friendliness [27]. The evaluation of SOP by a user-convenience questionnaire revealed very high ratings and the participants evaluated SOP as a very valuable tool for clinical routines. This finding is supported by previous publications showing that clinicians prefer SR over free text reporting [14,15,16,19,23,34,35,36]. Considering the fact that performing FESS requires a trainee surgeon to not only develop new manual dexterity skills, but also to possess a thorough anatomic understanding with spatial orientation [37,38,39], SOP may become a very valuable tool in the training of younger FESS surgeons by offering a systematic and standardized approach to operation planning [14].

The findings of the study are limited due to the small number of planning procedures. Every participant conducted only one COP and SOP; therefore, time to complete could have been shorter for both options after repeated accomplishment. Bias due to feedback from the template itself was minimized by scheduling COP before SOP. Nevertheless, there was a residual risk of bias due to learning or testing effects.

## 5. Conclusions

Implementation of a structured approach in operation planning may be a valuable tool in the training process of FESS. Further studies with a larger number of CT examinations are necessary to place the preliminary findings of our study within a broader perspective. In detail, a thorough investigation of the time expenditure required to produce SOP and, therefore, of its time-efficiency, in a population that was not previously trained in COP, will be of great interest to determine the impact of SOP on the learning process.

## Figures and Tables

**Figure 1 medicina-57-01062-f001:**
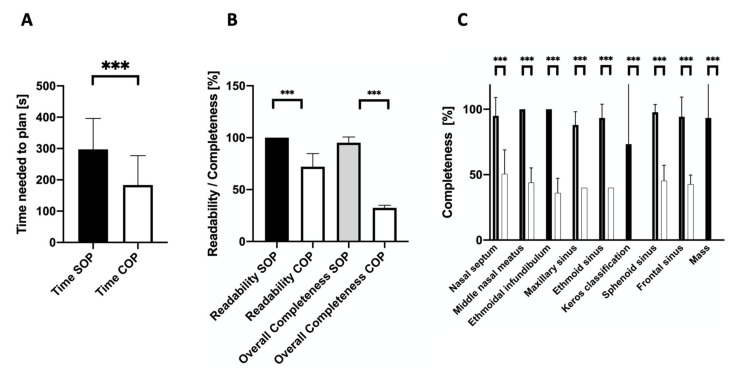
Time needed to plan (**A**), evaluation of legibility (**B**), and overall completeness of structured (SOP), as well as conventional operation planning (COP, (**B**,**C**)). Analysis of time expenditure for operation planning exhibits a significant difference in favor of conventional operation planning (COP). Using SOP results in significantly better legibility and overall completeness. In detail, all relevant anatomical features are addressed considerably more completely using SOP. *** *p* < 0.001.

**Figure 2 medicina-57-01062-f002:**
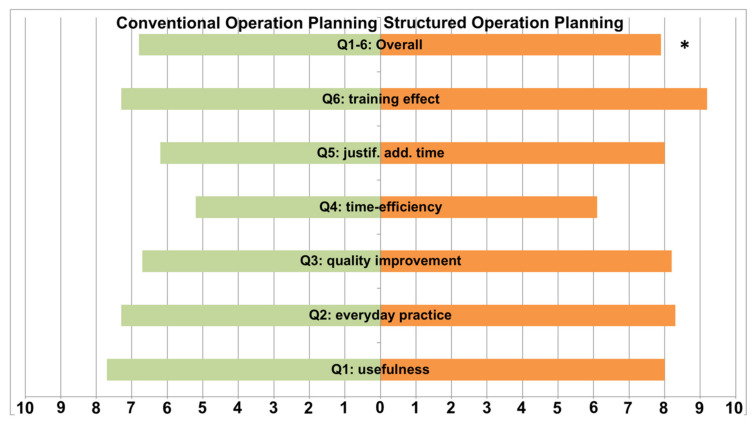
Evaluation of user convenience yields a significant overall preference for structured operation planning (SOP) by participating residents in comparison to conventional operation planning (COP). Detailed analysis showed a tendency in favor of SOP in all investigated items without reaching the level of significance. * *p* < 0.05.

**Table 1 medicina-57-01062-t001:** Demographic data and experience of the 15 participants included in the study. To evaluate individual levels of experience in FESS, a five-point scale (1: none; 5: very experienced) was utilized.

		Number of Participants	Percent%
All participants		15	100
Mean age, years (range)		33 (26–41)	
Gender			
Female		6	40
Male		9	60
Staus			
Resident		9	60
Consultant		6	40
Prior participation in a FESS course			
Yes		9	60
No		6	40
Work environment			
Privat practice		1	7
Municipal hospital		8	53
University medical center		6	60
Estimated level of Experience			
None	(1)	0	0
Little	(2)	2	13
Average	(3)	12	80
Experienced	(4)	1	7
Veryexperienced	(5)	0	0

## Data Availability

All the data are available from the corresponding author upon reasonable request.

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
