# Peer review of "ENT Residents Benefit from a Structured Operation Planning Approach in the Training of Functional Endoscopic Sinus Surgery"

_medicina, 2021, doi:10.3390/medicina57101062_

Round 1

Reviewer 1 Report

Generally the intro can be improved, the articles referring to the problem presented are quite new, but there are some more relevant. Methods used are appropriate. My question deals with inclusion and exclusion criteria: were they used. What problems were identified during the study. What are the weaknesses of the study design?

List of references can be improved:

Folia Morphologica 2021: A computed tomography comprehensive evaluation of the ostium of the sphenoid sinus and its clinical significance

  1. Jaworek-Troć, J. A. Walocha, J. Skrzat, J. Iwanaga, R. S. Tubbs, M. Mazur, M. Lipski, A. Curlej-Wądrzyk, T. Gładysz, R. Chrzan, A. Urbanik, M. P. Zarzecki

Ahead of Print

Folia Morphologica 2021: Extensive pneumatisation of the sphenoid bone — anatomical investigation of the recesses of the sphenoid sinuses and their clinical importance

  1. Jaworek-Troć, J. A. Walocha, M. Loukas, R. S. Tubbs, J. Iwanaga, J. Zawiliński, K. Brzegowy, J. J. Zarzecki, A. Curlej-Wądrzyk, E. Kucharska, F. Burdan, P. Janda, M. P. Zarzecki

Ahead of Print

Protrusion of the carotid canal into the sphenoid sinuses: evaluation before endonasal endoscopic sinus surgery

  1. Jaworek-Troć, J. A. Walocha, R. Chrzan, P. Żmuda, J. J. Zarzecki, A. Pękala, P. Depukat, E. Kucharska, M. Lipski, A. Curlej-Wądrzyk, M. P. Zarzecki

Folia Morphologica 2021 Vol 80, No 3 (2021)Translational Research in Anatomy

Volume 24, September 2021, 100126

Radio-anatomic variability in sphenoid sinus pneumatization with its relationship to adjacent anatomical structures and their impact upon reduction of complications following endonasal transsphenoidal surgeries

SolomonTesfaye Niguse Hambaa  Asfaw Gerbia Zenebe Negerib

https://doi.org/10.1016/j.tria.2021.100126

Translational Research in Anatomy

Volume 21, November 2020, 100079

Anatomical variations of the main septum of the sphenoidal sinus and its importance during transsphenoidal approaches to the sella turcica

Joanna Jaworek-Troć; Joe Iwanaga ; Robert Chrzan; Jacek J.Zarzecki  Paulina Å»mudaeAgataPÄ™kalaa Iwona M.Tomaszewskaf R. ShaneTubbsc JarosÅ‚awZawiliÅ„skia MichaÅ‚ P.Zarzeckia

https://doi.org/10.1016/j.tria.2020.100079

Author Response

Thank you very much for reviewing our article and for your valuable suggestions. The inclusion criteria were as follows: Subjects needed to be residents in otolaryngology with a basic knowledge in functional endoscopic sinus surgery (FESS) who participated in the 4 days annual course on FESS at the University Medical Center of Tübingen, Germany.

The exclusion criteria were as follows: No consent to participate in the study.

Problems concerning the study design did not arise during the conduction of the study. Problems concerning conventional operation planning were identified as low rates of consideration regarding specific intraoperative structures at risk which may pose potential risks intraoperatively. Additionally, user friendliness of conventional operation planning was considered to be significantly lower which may be a risk factor for insufficient compliance in clinical practice.

Weaknesses of the study design may include the number of participating residents, potential bias due to the chosen sequence of conventional (first) and structured (second) operation planning with potential training effects as well as various preexisting experience in FESS of the participants.

Reviewer 2 Report

Thank you for the opportunity to review this study. I find this article very interesting and well structured. Nonetheless, I cannot find the Table 2, even if it is mentioned in line 102. Furthermore, I think the Figure 2 (lines 118-119) should not be mentioned here, as probably the Figure 2 does not show these data (lines 115-118). Can the Authors show a copy of the SOP used?

Minor corrections:

  • Page 1, line 27, “COPs” is not previously defined
  • Page 1, line 25, “ can considered” probably needs “be”
  • Page 2, line 65, “SOP” should be defined here and not in line 68
  • Page 3, line 103, “Data of” probably is not completed
  • Lines 177, 189, 197, 207, the references should be sorted
  • Page 6, line 203, “highl” should be “high”

Author Response

Thank you very much for reviewing our article and for your valuable suggestions. Mentioning a Table 2 within the manuscript was typing error as it does not exist. Consequently, the text passage was deleted from the revised manuscript. The reference to Figure 2 in lines 115-119 was corrected to a reference to Figure 1, as the data mentioned in the text is shown there. A copy of the SOP used will be uploaded as Supplementary Figure 1. Minor corrections were carried out as suggested.
